# Fast Supraharmonic Estimation Algorithm Based on Simplified Compressed Sensing Model

**Zesen Gui, Qun Zhou \*, Hui Zhou, Zheng Liao and Ziyi Wang**

College of Electrical Engineering, Sichuan University, Chengdu 610065, China
\* Correspondence: zhouqun@scu.edu.cn; Tel.: +86-028-8540-5614

**Abstract:** The computational time of compressed sensing algorithms applied to supraharmonic needs to be improved in online applications. In this paper, a simplified supraharmonic compressive sensing model is proposed. The model first detects the supraharmonic raw spectral array to obtain the estimated sparsity and the index of supraharmonic emissions, which simplifies the sensing matrix in the iteration according to the index and then shortens the whole iteration time of compressed sensing. The simulation verifies that the model can reduce the computation time to less than half of the original compressed sensing model and does not affect the computation accuracy. Finally, the online application effect of the algorithm is verified by experiments.

**Keywords:** supraharmonic; compression sensing algorithm; sparsity; data distribution

## 1. Introduction

With the development of power systems, the distributed renewable energy sources (RES) connected to the distribution network is increasing [1,2]. On the grid side, a large number of inverters are connected to the grid, and high-frequency voltage and current harmonics are injected into the grid, too. On the user side, rectifiers, and chopper devices, such as switching the power supply and electric vehicle charging piles, are becoming increasingly popular. The current harmonics generated by these devices in a range from 2 kHz to 150 kHz also affect the distribution network and adjacent devices [3]. The voltage and current harmonics mentioned above in the range from 2 kHz to 150 kHz are collectively referred to as supraharmonic [4].

Power electronics and other supraharmonic emission sources directly affect the normal operation and service life of nearby equipment. In the papers [5,6], it was found through practical measurements that the high-frequency voltage distortion caused by supraharmonics leads to faults and noise in household appliances. The device capacitors also heat up due to excessive supraharmonic currents, which further affect the device's life [7]. In addition to this, supraharmonics affect the power quality of the distribution network and power line communication [8]. The meter causes significant errors when subjected to supraharmonic interference [9]. Supraharmonics affect the quality of power supply to the grid. The literature [10,11] measured the supraharmonic emission generated by EV charging to further investigate the effect of related devices on the supply voltage. The paper [12] studied the effect of grid impedance on the propagation of supraharmonics, and experimentally summarized some laws on the interaction between supraharmonics and the grid.

For the above ongoing research, fast and accurate supraharmonic measurement methods are indispensable, but mature measurement methods for supraharmonic identification and estimation are still lacking [13]. The measurement standards IEC 61000-4-7 [14], IEC 61000-4-30 [15], and CISPR 16-2-1 [16] propose three supraharmonic detection methods. The literature [13,17] compared the existing standards, the CISPR 16-2-1 method, and the IEC 61000-4-7 method, which have a frequency resolution

of 200 Hz, and the detection amplitude is the same in most cases. However, the amount of data processed by these two methods is more than 10 times that of the IEC 61000-4-30 method. The weakness of the IEC 61000-4-30 method is that the frequency resolution is 2 kHz and the method only analyses 8% of the measurement data.

The existing standard methods for supraharmonics meet the needs of different application situations, but in order to have a standardized method for the online detection of supraharmonics, it is necessary to improve the computational complexity, computational time, and computational accuracy. According to the standard CISPR 16-2-1, the measurement bandwidth of the frequency band A (9–150 kHz) is 200 Hz, so it takes 1410 measurements in this band, and the measuring time is more than 2 min. To reduce the computation time, the literature [18] was improved based on CISPR 16-2-1 by using a cascaded phase-locked loop to obtain the higher energy supraharmonic components instead of the swept frequency detection in the original standard. The detection time is significantly reduced, but it still takes a few seconds to complete. In order to reduce the computational burden of CISPR 16-2-1, a new digital quasi-peak detection method was proposed in the literature [19], which indirectly reduced the computation time. Although the methods of calculation based on CISPR 16-2-1 have been improved, the computing time still takes over 200 ms, which is not suitable for online measurements of supraharmonics.

IEC 61000-4-30 and IEC 61000-4-7 use DFT for spectrum analysis. According to IEC 61000-4-30, power quality analysis requires 200 ms time series data collection. Harmonics below 2 kHz are typically sampled using a sampling rate of 10 kSP/s, so the amount of data for harmonic analysis is approximately 2000. For a supraharmonic of 2-150 kHz, a minimum sampling rate of 300 kSP/s is required, and the minimum amount of data to be analyzed is 60,000, which is 30 times higher than that of the harmonics. The increased data processing requirements put a higher demand on the computational resources of the power-quality instruments used for online measurements in the field. Faced with a large amount of data, IEC 61000-4-7 performed spectrum analysis on all the data, and IEC 61000-4-30 retained only 8% of the data for spectrum analysis. Therefore, the frequency resolution and signal coverage of the IEC 61000-4-30 supraharmonic measurement method is lower than that of other standard methods.

In order to balance the computation time and frequency resolution, a supraharmonic compression sensing model was proposed in the literature [20,21] to detect supraharmonics using the CS-OMP algorithm to recover the detection results with a 200 Hz frequency resolution from the 0.5 ms duration data. The literature [20] further proposed the MCS for supraharmonics, which reduces the number of calculations from N to 1 for N sets of measurements using the combined sparsity of high-resolution spectral arrays. This method can calculate all signals in the IEC 61000-4-30 method. After that, the literature [22] achieved adaptive sparsity by comparing iterative residuals, which reduced the computational error but significantly increased the computation time. The literature [23] used the Bayesian algorithm CS-BCS instead of the CS-OMP greedy algorithm to optimize the detection performance and frequency detection error of intermittent supraharmonic emissions yet increased the overall reporting delay. The literature [24] uses CS-OMP to select the supraharmonic emission frequencies and compute TFT expansion at the fundamental frequency and at these frequencies to achieve the more accurate detection of time-varying signals. The above study further extends the application prospects of the compression sensing algorithm in the online measurement of supraharmonics. However, the existing compressed sensing model needs to determine the number of iterations according to the sparsity (or residuals) and then calculate the matrix index of the supraharmonic vectors based on the sensing matrix. In the iterative process, the inner product of any column of the sensing matrix with the current residual needs to be computed, and the computation time grows linearly with increasing sparsity. Therefore, this paper proposes a method to estimate the sparsity based on the probability density model and, in the process, obtains the supraharmonic spectrum index, simplifies the sensing matrix based on the spectrum index, and then simplifies the compressed sensing model. The model only needs to calculate the

inner product of any column in the simplified sensing matrix with the current residuals in a single iteration, thus reducing the iteration time and resulting in a reduction of the computational complexity of the supraharmonic compressed sensing algorithm.

This paper is organized as follows. Section 2 introduces the principle and computational process of the simplified supraharmonic compression sensing model; Section 3 investigates the computational complexity and computational accuracy of the algorithm; Section 4 verifies the computational time and accuracy of the algorithm through experiments, and finally reviews the contributions of this paper and makes conclusions.

## 2. Algorithm Theory

### 2.1. Compressed Sensing Models for Supraharmonic

The sampled data of 200 ms power signals containing supraharmonics are filtered by a 2–150 kHz bandpass filter, and the data are evenly divided into 400 groups of $\Delta T = 0.5\text{ms}$ small data blocks, among which the supraharmonic components contained in the time-domain data of any small data block can be expressed as:

$$
\begin{aligned}
\text{s}(n) &= \sum_{k=1}^{N} A_k \cos(2\pi f_k n T_s + \varphi_k) \ n \in (0, \cdots, N-1) \\
&= \sum_{k=1}^{N} \left( \frac{A_k}{2} e^{j(\varphi_k + 2\pi f_k n T_s)} + \frac{A_k}{2} e^{-j(\varphi_k + 2\pi f_k n T_s)} \right)
\end{aligned}
\tag{1}
$$

where $A_k, f_k,$ and $\varphi_k$ are the amplitude, frequency, and initial phase of each frequency component, respectively. Assuming that the parameters are stable, $T_s$ is the sampling interval, $f_s$ is the reciprocal of the sampling frequency, and the length of the sequence is $N = f_s \Delta T$.

After $s(n)$ processing with DFT, it can be expressed as:

$$
x(k) = \frac{1}{N} \sum_{n=0}^{N-1} s(n) e^{-j2\pi kn/N}
\tag{2}
$$

where $1 \leq k \leq N$, the frequency resolution $\Delta f = f_s/N$ is 2 kHz, neglecting the effect of the negative frequency points in Equation (1). Substituting Equation (1) into Equation (2) yields:

$$
x(k) = \frac{1}{2} \sum_{k=1}^{N} \left( A_k e^{j\varphi_k} \frac{1}{N} \sum_{n=0}^{N-1} e^{-j2\pi n(k/N - f_k T_s)} \right)
\tag{3}
$$

where the sum of the equiprobable series $\frac{1}{N} \sum_{n=0}^{N-1} e^{-j2\pi n(k/N - f_k T_s)}$ is calculated as:

$$
\begin{aligned}
&\frac{1}{N} \frac{1 - e^{-j2\pi N(k/N - f_k T_s)}}{1 - e^{-j2\pi(k/N - f_k T_s)}} \\
&= \frac{\sin \pi N(k/N - f_k T_s)}{N \sin \pi(k/N - f_k T_s)} e^{-j\pi(N-1)(k/N - f_k T_s)} \\
&= D_N \left( \frac{k}{N} - f_k T_s \right)
\end{aligned}
\tag{4}
$$

The final expression of the original spectral array can be obtained as:

$$
x(k) = \frac{1}{2} A_k e^{j\varphi_k} \sum_{k=1}^{N} D_N \left( \frac{k}{N} - f_k T_s \right)
\tag{5}
$$

To improve the frequency resolution, an interpolation factor $F$ is introduced. Thus, the frequency resolution can be refined to $\Delta' f = \Delta f / F$, and the total spectral lines are $N' = NF$. Thus:

$$
f_k T_s \approx \frac{r}{N\prime}
\tag{6}
$$

where $r$ is the $r$ spectral line in the new frequency resolution. The new original spectral array is:

$$x(k) \simeq \frac{1}{2} \sum_{r=1}^{N\prime} D_N\left(\frac{k}{N} - \frac{r}{N\prime}\right) A_r e^{j\theta_r} \tag{7}$$

The matrix form of Equation (7) is expressed as:

$$\mathbf{x}_k \simeq \frac{1}{2}\mathbf{DA} \tag{8}$$

that is

$$\begin{pmatrix} x(1) \\ \vdots \\ x(k) \\ \vdots \\ x(N) \end{pmatrix} \simeq \frac{1}{2} \begin{pmatrix} D_{N(1,1)} & \cdots & D_{N(1,r)} & \cdots & D_{N(1,N')} \\ \vdots & \ddots & \vdots & \vdots & \vdots \\ D_{N(k,1)} & \cdots & D_{N(k,r)} & \cdots & D_{N(k,N')} \\ \vdots & \vdots & \vdots & \ddots & \vdots \\ D_{N(N,1)} & \cdots & D_{N(N,r)} & \cdots & D_{N(N,N')} \end{pmatrix} \begin{pmatrix} A_1 e^{j\varphi_1} \\ \vdots \\ A_r e^{j\varphi_r} \\ \vdots \\ A_{N'} e^{j\varphi_{N'}} \end{pmatrix} \tag{9}$$

where the elements of the observation matrix $\mathbf{D}$ are $D_{N(k,r)} = D_N(k/N - r/N')$. From Equation (9), the interpolated estimation matrix $\mathbf{A}$ can be solved when the original spectral array $\mathbf{x}_k$ and the sensing matrix $\mathbf{D}$ have been obtained, but the huge amount of data and the inverse of the observation matrix are difficult to achieve and the error rate is not low, so the direct solution rarely occurs in online applications, and the compressed sensing algorithm is needed to further identify the supraharmonic in the estimation matrix.

### 2.2. Simplified Compressed Sensing Model and its Calculation

Existing supraharmonic compression sensing algorithms require a given sparsity to control the number of supraharmonic spectra detected by the algorithm before calculation, or the maximum calculation residual is set to control the calculation accuracy and indirectly control the detection results; these methods are more suitable for detecting known supraharmonic emissions. However, if an unknown signal is actually measured, the sparsity of the algorithm needs to be estimated based on the test signal so that the calculation result is closer to the supraharmonic emission of the unknown signal.

#### 2.2.1. The Prediction of Sparsity S

It is most convenient to estimate sparsity $S$ directly from the original spectral array $\mathbf{x}_k$, provided that the original spectral array $\mathbf{x}_k$ is known and no additional computational effort is added. The current supraharmonic test results contain a large number of noise spectra in addition to supraharmonic emissions, while the original spectral array contains only amplitude and phase information, and the phase information cannot discriminate the supraharmonic and noise in the original spectral array and can only be used to discriminate the supraharmonic emission by setting the amplitude threshold.

The ideal amplitude threshold should be slightly higher than the noise amplitude in order to detect all the supraharmonic emissions as much as possible. While the arbitrary raw spectral array amplitude information can be divided into normal and skewed distributions according to the probability density distribution, where the normal distribution usually uses the mean to represent the average level of the data while adjusting the amplitude threshold with the standard deviation based on the mean in order to be close to the ideal threshold, the skewed distribution uses the median and median deviation to calculate the amplitude threshold. First, calculate the skewness entropy of the original spectral array data [25]:

$$\beta_1 = \frac{1}{N} \sum_{k=1}^{N} (x_k - \overline{x})^3 \Big/ \left(\frac{1}{N} \sum_{k=1}^{N} (x_k - \overline{x})^2\right)^{3/2}, \tag{10}$$

where $\mathbf{x}_k$ represents the spectral value of the original spectral array, $\overline{x} = \frac{1}{N}\sum_{k=1}^{N} x_k$ and then calculate the raw spectral array data kurtosis entropy:

$$\beta_2 = \frac{1}{N}\sum_{k=1}^{N}(x_k - \overline{x})^4 \Big/ \left(\frac{1}{N}\sum_{k=1}^{N}(x_k - \overline{x})^2\right)^2, \tag{11}$$

before finally using the Jarque-Bera test:

$$JB = \frac{N}{6}\left[\beta_1^2 + \frac{(\beta_2 - 3)^2}{4}\right], \tag{12}$$

to determine whether the original spectral array belongs to normal or skewed distribution [26]. For the normal distribution model, the amplitude threshold of the original spectral array $x_{Th}$ is set to [27,28]:

$$x_{Th} = \overline{x} + \beta_1 \times \sigma(x_k) \tag{13}$$

where $\sigma(x_k) = \sqrt{\frac{1}{N}\sum_{k=1}^{N}(x_k - \overline{x})^2}$; for skewed distributions:

$$x_{Th} = \overline{x} \tag{14}$$

Comparing the original spectral array and the amplitude threshold, a matrix consisting of the number of spectral lines greater than the amplitude threshold and the corresponding matrix index can be obtained:

$$\mathbf{I} = \{k_1, k_2, \cdots, k_S\} \tag{15}$$

where $k$ represents the index of the supraharmonic emission band selected from $k$ in Equation (7). The size of the matrix I represent the number of supraharmonic emissions in the original spectral array, so it is regarded as the sparsity in the proposed algorithm.

### 2.2.2. Simplified Compressed Sensing Model

After estimating the sparsity and the matrix index, it is already possible to predict the supraharmonic emission bands, and only a more accurate emission band needs to be calculated by the compressed sensing algorithm. The computational complexity of the compressed sensing algorithm mainly comes from the $N \times N'$ dimensional observation matrix $\mathbf{D}$ in Equation (8), and the observation matrix $\mathbf{D}$ can be simplified accordingly after obtaining the matrix index $\mathbf{I}$. The observation matrix $\mathbf{D}$ consists of the $N'$ column vector, and the column vector that can predict the supraharmonic emission has the following relationship with the matrix index $\mathbf{I}$:

$$\mathbf{D}_I = \begin{pmatrix} D_{N(k,i_1 \times F)} & D_{N(k,i_2 \times F)} & \cdots & D_{N(k,i_S \times F)} \end{pmatrix} \tag{16}$$

However, the index $\mathbf{I}$ represent the 2 kHz bandwidth and the column vector in the 2 kHz/$F$ bandwidth, so we need to extend the range of $\mathbf{D}_I$:

$$\mathbf{D}_I = \begin{pmatrix} \cdots & D_{N(k,i \times F - F + 1)} & \cdots & D_{N(k,i \times F)} & \cdots \end{pmatrix} \tag{17}$$

The expanded $\mathbf{D}_I$ is a $N \times (F \cdot S)$ dimensional matrix, and since $F \cdot S \ll N'$, the solution complexity of the compressed perceptual model using $\mathbf{D}_I$ should be lower than that of the model containing the $N \times N'$ dimensional matrix $\mathbf{D}$.

Therefore, Equation (8) can be simplified to a new compressed perceptual model:

$$\mathbf{x}_k \simeq \frac{1}{2}\mathbf{D}_I\mathbf{A}_I \tag{18}$$

where $\mathbf{A}_I$ is the estimated result in the supraharmonic emission band. In order to make $\mathbf{A}_I$ directly the result of spectrum analysis, the size of $\mathbf{A}_I$ cannot be changed. Therefore, the

column vectors in $\mathbf{D}_I$ remain unchanged, the other columns were filled with zeros, and the size of the matrix $\mathbf{D}_I$ remained as $N \times N'$.

### 2.2.3. Compression Sensing Estimation of Spectrum

The supraharmonic spectral array represented by Equation (18) is further filtered using the OMP algorithm, and the algorithm iteration process is as follows.

Algorithm input parameters: $N \times 1$ dimensional original spectrum array $\mathbf{x}_k$; $N \times (F \cdot S)$ dimensional observation matrix $\mathbf{D}_I$; sparsity (number of cycles) $S$.

Algorithm parameter initialization. The set residuals are $\mathbf{r}_0 = \mathbf{x}_k$, the index set is $\Lambda_0 = \Phi$, and the sub-matrix of $\mathbf{D}_I$ is $\mathbf{D}_\Lambda = []$, number of iterations: $t = 1$.

Algorithm iteration process: Calculate the inner product of the current residual $\mathbf{r}_{t-1}$ and the column vector of the observation matrix $\mathbf{D}_I$. Solve for the position index $\lambda_t$ of the column vector when the inner product is at a maximum.

$$\lambda_t = \arg\max \left| \langle \mathbf{D}_{r_{Th}}, \mathbf{r}_{t-1} \rangle \right| i = 1, \cdots, FK_{Th} \tag{19}$$

Expanding the index set: $\Lambda_t = \Lambda_{t-1} \cup \{\lambda_t\}$ and sub-matrix $\mathbf{D}_\Lambda = [\mathbf{D}_\Lambda \ \mathbf{D}_{\lambda_t}]$. Set to zero the $\lambda_t$ column vector of $\mathbf{D}_I$: $\mathbf{D}(:, \lambda_t) = 0$. Calculate a new estimate vector:

$$\hat{\mathbf{u}}_t = \arg\min \|\mathbf{v} - \mathbf{D}_\Lambda \mathbf{u}_t\|_2^2 = \left( \mathbf{D}_\Lambda^H \mathbf{D}_\Lambda \right)^{-1} \mathbf{D}_\Lambda^H \mathbf{v} \tag{20}$$

Update the residuals $\mathbf{r}_t = \mathbf{x}_k - \mathbf{D}_\Lambda \mathbf{u}_t$. If $t > K$ and stop the loop; otherwise, if $t = t + 1$, continue iterating.

The output of the algorithm: the index support set is $\Lambda = \Lambda_t$ and the perceptual sub-matrix $\mathbf{D}_\Lambda$. After recovering the support set $\Lambda$ and sub-matrix $\mathbf{D}_\Lambda$, the high-resolution spectrum array can be recovered using least squares:

$$\hat{\mathbf{A}}_I = \arg\min \|\mathbf{x}_k \text{-} \mathbf{D}_\Lambda \mathbf{A}_I\|_2^2 = \left( \mathbf{D}_\Lambda^H \mathbf{D}_\Lambda \right)^{-1} \mathbf{D}_\Lambda^H \mathbf{x}_k \tag{21}$$

where $\hat{\mathbf{A}}_I$ includes all the screened high-resolution spectra. Each line corresponds to the phase volume of each frequency component at different times. The frequency, amplitude, and phase matrix of the main supraharmonic emissions can be obtained by:

$$\mathbf{f} = (\Lambda - 1) \times \Delta' f \tag{22}$$

$$\mathbf{A} = \text{abs}\left( \hat{\mathbf{A}}_I \right) \tag{23}$$

$$\theta = \text{angle}\left( \hat{\mathbf{A}}_I \right) \tag{24}$$

### 2.3. Discussion of Relevant Parameters

### 2.3.1. Original Spectral Array of Multi-measurement Vectors

The original spectral array of multi-measurement vectors has joint sparsity [20], so the original spectral array of M for the single-measurement vectors is combined as:

$$\mathbf{S} = [\mathbf{x}_{k,1}, \cdots, \mathbf{x}_{k,m}, \cdots, \mathbf{x}_{k,M}] \tag{25}$$

Calculate the autocorrelation matrix $\mathbf{R}_S$ of $\mathbf{S}$:

$$\mathbf{R}_S = E[\mathbf{S}\mathbf{S}^H] \tag{26}$$

where the superscript $^H$ indicates the transpose and complex conjugate. Then, decompose the $\mathbf{R}_S$ eigenvalue:

$$\mathbf{R}_S = \mathbf{V}_S \Lambda_S \mathbf{V}_S^H \tag{27}$$

where $\Lambda_S$ and $\mathbf{V}_S$ are the eigenvalue matrix and eigenvector matrix, respectively, so that $\mathbf{d}_K = \sqrt{diag(\Lambda_S)}$.

The eigenvector matrix $\mathbf{V}_S$ is multiplied by $\mathbf{d}_K$ to obtain the joint raw spectral array of the multi-measurement vectors $\mathbf{v}$:

$$\mathbf{v} = \mathbf{V}_S \mathbf{d}_K \tag{28}$$

Instead of the original spectral array $\mathbf{x}_k$, the multi-measurement vectors array $\mathbf{v}$ can be filtered to recover the support set $\Lambda$ and the perceptual submatrix $\mathbf{D}_Y$ of the supraharmonic emission, and then the spectrum is recovered by the least squares method.

### 2.3.2. The Relationship between the Interpolation Factor F and the Sparsity S

From the literature, it is known that the interpolation factor F and the sparsity S have the following relationship:

$$F \leq \frac{1}{N} \exp(N/mS) \tag{29}$$

where $m \in (0, 1]$ in the application $m = 1$, therefore, $F \leq \frac{1}{N} \exp(N/S)$. The interpolation factor $F$ is too small to achieve the effect of improving the frequency resolution, and when $F$ becomes large, the calculation complexity and algorithm recovery difficulty also increase. Therefore, $F$ finally takes the value of 10, at which time the frequency resolution is 200 Hz for the frequency resolution of the supraharmonic measurement proposed in IEC61000-4-7, $S \leq N/\ln(10 \times N)$.

### 3. Algorithm Simulation Analysis

In this section, the performance of the proposed algorithm is verified by MATLAB. The orthogonal matching tracking algorithm (CS-OMP), Bayesian algorithm (CS-BCS), and CS-TFM algorithms are selected for comparison. The performance of the proposed algorithms was evaluated by analyzing the computation time and estimation accuracy of each algorithm comprehensively. The simulation program of each algorithm was run on MATLAB R2018b with a computer configuration of a 3.20 GHz AMD Ryzen 7 5800H processor, 16-GB memory, and a Windows 11 64-bit operating system. In this paper, two metrics of frequency error (FE) and magnitude relative error (ME) are introduced to evaluate the algorithm detection effectiveness [17].

$$FE = \left| f_{test} - f_{ref} \right| \tag{30}$$

$$ME = \left| M_{test}/M_{ref} - 1 \right| * 100 \tag{31}$$

$M_{test}$ and $f_{test}$ indicate the test signal amplitude and frequency, respectively, and $M_{ref}$ and $f_{ref}$ indicate the reference signal amplitude and frequency.

### 3.1. Simulation Model

In order to verify the performance of the algorithm, this paper tested using a single-frequency signal and the Class D test signal proposed in the literature [16]. The signal model of the test signal is shown below:

$$s(n) = \sum_{k=1}^{N} A_k \cos(2\pi f_k n T_s + \varphi_k) \tag{32}$$

where $A_k$, $f_k$, and $\varphi_k$ are the amplitude, frequency, and initial phase of the frequency component of the test signal. The specific parameters of the frequency and amplitude of the single-frequency signal are shown in Table 1. The constant amplitude single-frequency signal was used to evaluate the amplitude accuracy of a specific frequency and also to simulate the supraharmonic emission of some LEDs. The Class D test signal proposed in the literature [17] uses a supraharmonic distortion with a frequency representative of the

actual grid, containing narrowband supraharmonic, broadband supraharmonic, power line communication supraharmonic, and environmental noise, which can more realistically represent the suprahamonic distortion in the grid. Some of the frequencies and amplitudes in the Class D test signal are adjusted in this paper; the final test signal spectrum used is shown in Figure 1, and the peak narrowband supraharmonic emissions are summarized in Table 2 for the convenience of comparing the test results. The phase of the test signal is set to a certain value in the range. Both the single-frequency signal and the Class D test signal simulate a data length of 200 ms sampled at a sampling rate of 500 KSP/s, i.e., the data from 100 k sampling points with a time interval of 2μs were analyzed.

**Table 1.** Frequency and amplitude of single frequency test signals.

| No. | Frequency (kHz) | Amplitude (mV) |
|-----|-----------------|----------------|
| 1 | 49.8 | 500 |
| 2 | 50.2 | 750 |
| 3 | 99.8 | 800 |
| 4 | 100.2 | 1000 |

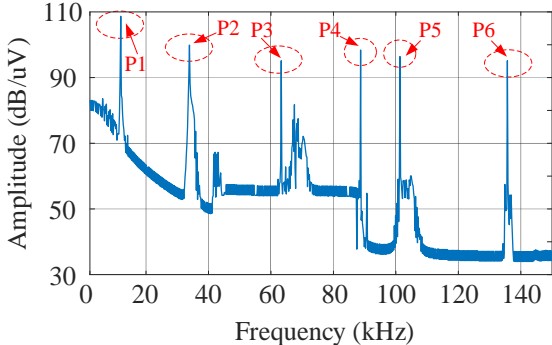

**Figure 1.** Class D test signal spectrum.

**Table 2.** Class D test signal main emission spectrum.

| No. | Frequency (kHz) | Amplitude (mV) |
|-----|-----------------|----------------|
| 1 | 12.2 | 271.09 |
| 2 | 38.8 | 97.83 |
| 3 | 63.4 | 57.33 |
| 4 | 88.8 | 83.09 |
| 5 | 101.4 | 66.82 |
| 6 | 135.8 | 57.09 |

*3.2. Algorithm Complexity and Computation Time*

The computational burden of the original compressed sensing model mainly comes from calculating the current residuals and the inner product of the column vectors of the sensing matrix during the iterative process, while the simplified compressed sensing model (Equation (18)) proposed in this paper mainly reduces the computational complexity and shortens the computation time by decreasing the dimension of the sensing matrix. In order to demonstrate the simplification effect more intuitively, the CS-OMP algorithm is used as an example to compare the differences in computational complexity between the two models. In order to make the calculation time of the simulation more valuable, the calculation time in the following table is the average of 200 simulation times.

Under the simulation condition of a 500 KSa/s sampling rate, the length N of the timing signal is 250, and the original spectral array also contains 250 spectral components. When the interpolation factor F is 10, the number of spectral components N' recovered by compression Sensing should be 2500. Therefore, when the sparsity is *S*, the number of

times the original compressed Sensing model uses the OMP method to calculate the current residuals and the inner product of the column vectors of the Sensing matrix should be $2500 \times S$. The number of calculations after simplification is $10 \times S^2$, and the supraharmonic recovery spectral array satisfies the requirement of algorithm sparsity, i.e., $10 \times S < 2500$, so the number of times the simplified compressed sensing matrix calculates the current residuals and the inner product of the column vectors of the sensing matrix should be smaller than the original compressed Sensing matrix model. Figure 2 shows the relationship between the number of times the two models calculate the current residuals, the inner product of the column vectors of the perceptual matrix, and the sparsity $S$. As the sparsity $S$ increases, the difference between the number of times the two models correspond to the technique increases, and the ratio between the number of times the original model calculates and the number of times the simplified model calculates decreases, but even when the sparsity $S$ is taken as 50, the number of times the simplified model calculates is still 1/5 of the original model.

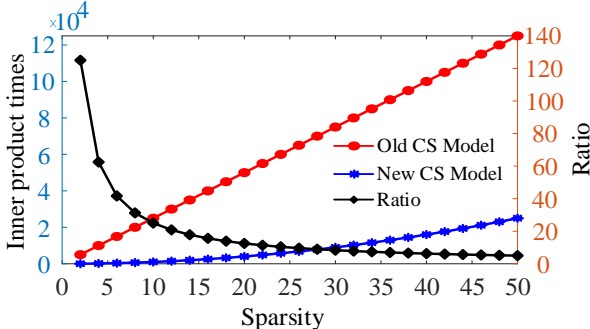

**Figure 2.** Comparison of the number of times to calculate the residuals and the inner product of the column vectors of the sensing matrix for two compressed sensing models.

Then, the effect of the simplified model on the computation time of the algorithms was verified, and the computation time of each algorithm is shown in Table 3 for a single frequency test signal. The computation time of the MCS-OMP algorithm depends on the size of the sparsity, but the simulations and experiments in the literature [20] were performed using a given sparsity with a known number of major supraharmonic emissions of the signal so that the CS-TFM algorithm used a time of 12–13 ms for a given sparsity of two. While the MCS-OMP algorithm uses a multi-measurement vector model, so the computation time increases, the MCS-BCS has a longer computation time of 14–15 ms than the MCS-OMP. The sparsity estimated by the proposed algorithm also takes the value of two, and the maximum computation time is 6 ms, which has a shorter computation time with the same value of sparsity as the MCS-OMP algorithm. Based on the sparsity of spectrum estimation, the proposed algorithm simplifies the sensing matrix to obtain Equation (18). For single-frequency test signals with small sparsity, the time reduction is not yet obvious, so the effect of the compressed sensing model obtained by Equation (18) on reducing the computation time was further verified by the class D test signals.

**Table 3.** Single frequency test signal calculation time.

| Algorithm | Time (ms) |
| --- | --- |
| new proposal | 5.7 |
| MCS-OMP | 13.4 |
| MCS-BCS | 14.2 |
| CS-TFM | 12.1 |

Figure 3 compares the computation time of the simplified compressed sensing model proposed in this paper and the MCS-OMP model in the literature for Class D test signals using a given sparsity without using the probability density distribution model to estimate

the sparsity, and the computation time is taken as the average of 10 simulations. At the minimum sparsity of two, the computation time was the same as that in Table 3, while the difference in computation time became more obvious as the sparsity increased. At the sparsity of 50, the computation time of the proposed algorithm was 59 ms, and the computation times of MCS-OMP, MCS-BCS, and CS-TFM were 178.4 ms, 212.296 ms, and 188.752 ms, respectively, and the proposed algorithm reduced the computation time of the MCS-OMP algorithm by 2/3. In the supraharmonic bandwidth, with a 200 Hz frequency resolution used by the proposed algorithm, 740 spectral bins were obtained, which corresponded to a sparsity range of 1480. It is only necessary to calculate all 740 spectrums if there are supraharmonic emissions in all bands. However, the number of actually measured supraharmonic emissions obtained in the current study was much smaller than 740 [10,11]. Therefore, the simplified compressed perceptual model represented by Equation 18 can effectively reduce the computation time of the MCS-OMP model, and the larger the value of sparsity taken, the more obvious the computation time reduction effect is.

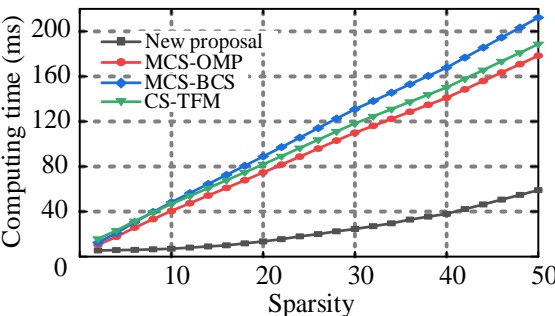

**Figure 3.** Comparison of calculation time of different algorithms for Class D test signals.

The computation time trend of the same compressed sensing model in Figure 3 is basically the same as the trend of the number of inner product calculations in Figure 2, which indicates that the number of inner product calculations is the main influence on the computation time.

The estimated sparsity of the Class D test signal using the probability density distribution model is 40, and the total computation time is 43 ms. Comparing the computation time of the directly given sparsity in Figure 2, the estimated sparsity increases the computation time by 5 ms, but the computation time of the MCS-OMP algorithm is 141 ms at the given sparsity of 40. Even if the time of the estimated sparsity is added, the computation time of the algorithm in this paper is still within half of the time of the MCS-OMP algorithm. Although the estimated sparsity increases the computation time, the overall computation time can be reduced more by simplifying the compressed perceptual model.

### 3.3. Calculation Accuracy

The results of the frequency and amplitude estimation for single-frequency test signals and Class D test signals are shown in Tables 4 and 5. From the two tables, it can be seen that the frequency estimation error FE for all algorithms at the main supraharmonic emission is 0. The amplitude estimation accuracy of the proposed algorithms is basically the same as that of CS-OMP, with the relative error ME within 0.2. The estimation accuracy of CS-BCS and CS-TFM is relatively higher, with the relative error of ME within 0.1, but there are still errors with the actual values.

**Table 4.** Single frequency test signal amplitude test.

| No. | New Proposal | | CS-OMP | | CS-BCS | | CS-TFM | |
|---|---|---|---|---|---|---|---|---|
| | FE | ME | FE | ME | FE | ME | FE | ME |
| 1 | 0 | 0.15 | 0 | 0.13 | 0 | 0.07 | 0 | 0.04 |
| 2 | 0 | 0.13 | 0 | 0.14 | 0 | 0.09 | 0 | 0.06 |
| 3 | 0 | 0.17 | 0 | 0.18 | 0 | 0.1 | 0 | 0.07 |
| 4 | 0 | 0.19 | 0 | 0.2 | 0 | 0.1 | 0 | 0.09 |
| 5 | 0 | 0.15 | 0 | 0.13 | 0 | 0.07 | 0 | 0.04 |
| 6 | 0 | 0.13 | 0 | 0.14 | 0 | 0.09 | 0 | 0.06 |

**Table 5.** Class D test signal amplitude test.

| No. | New Proposal | | CS-OMP | | CS-BCS | | CS-TFM | |
|---|---|---|---|---|---|---|---|---|
| | FE | ME | FE | ME | FE | ME | FE | ME |
| 1 | 0 | 0.12 | 0 | 0.13 | 0 | 0.07 | 0 | 0.03 |
| 2 | 0 | 0.13 | 0 | 0.14 | 0 | 0.08 | 0 | 0.05 |
| 3 | 0 | 0.13 | 0 | 0.14 | 0 | 0.09 | 0 | 0.06 |
| 4 | 0 | 0.15 | 0 | 0.15 | 0 | 0.09 | 0 | 0.07 |
| 5 | 0 | 0.16 | 0 | 0.17 | 0 | 0.09 | 0 | 0.08 |
| 6 | 0 | 0.17 | 0 | 0.17 | 0 | 0.1 | 0 | 0.09 |

To further verify the robustness of the algorithm and the application effect under the MMV model, Class D test signals are still used to make the estimation results of the amplitude and frequency of the algorithm in this paper at different times. Figure 4 gives the amplitude variation in the main supraharmonic emission of the Class D test signal within 200 ms.

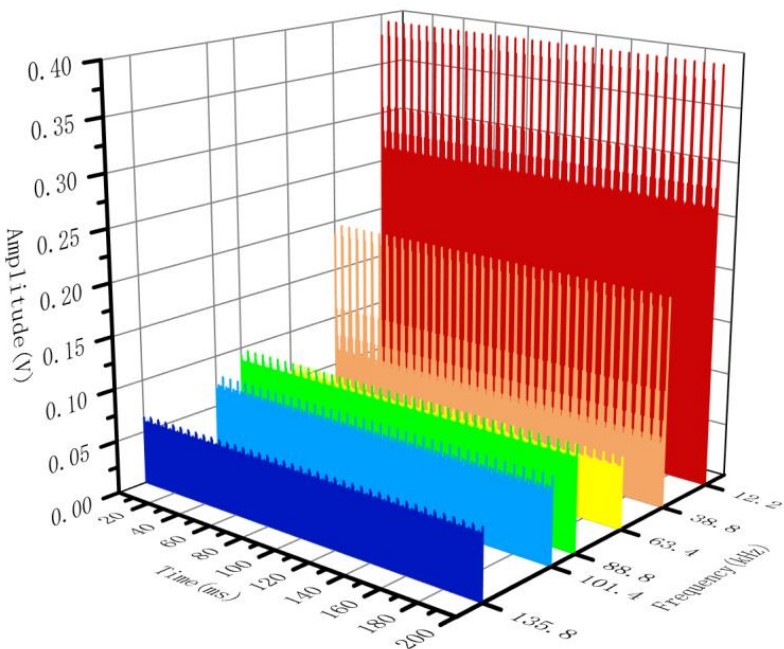

**Figure 4.** Class D test signal simulation results.

### 4. Experimental Test

In this section, the proposed algorithm is tested experimentally. The voltage probe used for the experiment is RIGOL RP1025D; the current probe used is RIGOL RP1004. The signals acquired by the voltage and current probes are passed through the third-order elliptic high-pass filter proposed in the literature [29], and then the data acquisition and processing are realized through the NI 9223 voltage acquisition module and the NI 9057

control box. The sampling frequency is set to 500 kHz and the sampling time is 200 ms. In the following, the computation time and computation accuracy of the algorithm of this paper in NI 9057 is examined using LED lights and Class D test signals, respectively, and the computation time of the MCS-OMP algorithm is added for comparison.

### 4.1. Class D Test Signal Experiment

As in Figure 5, the waveform of the Class D test signal is stored in a signal generator for continuous output and is then amplified by a power amplifier with a supraharmonic emission source with a 20-ohm power resistor as the load. The voltage probe is connected to both ends of the load. The experimental voltage waveform is shown in Figure 6.

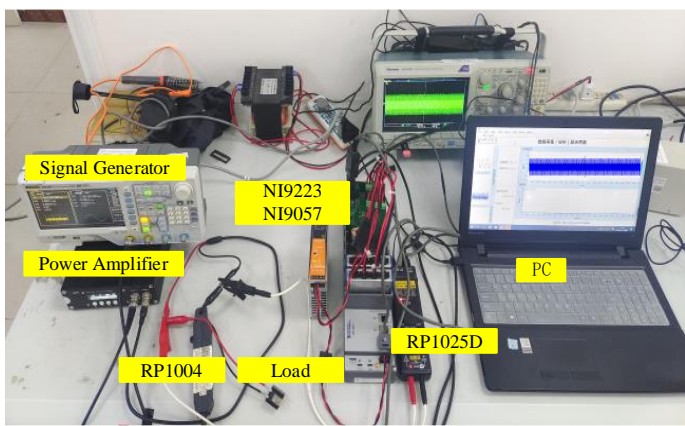

**Figure 5.** Class D signal test settings.

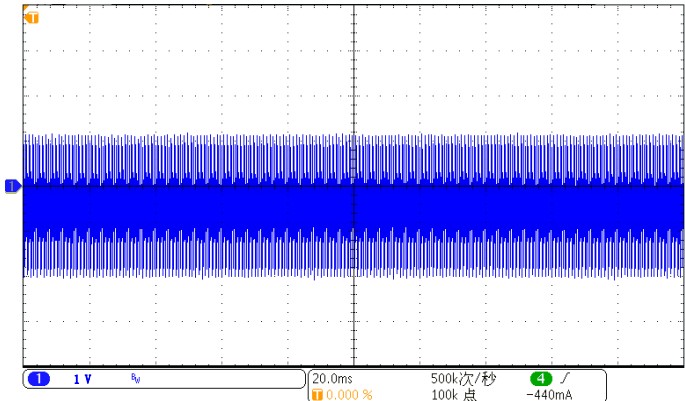

**Figure 6.** Class D signal supraharmonic voltage waveform.

The voltage sparsity predictions are 40, and a total of 20 supraharmonic emissions were detected in the supraharmonic band, containing the six major emissions in Table 2. The test results are shown in Figure 7. The detected six main supraharmonic emissions have the same frequency as the set frequency, the amplitude shows periodic fluctuations, and the relative error between the average value and the setting amplitude is within 0.2%.

### 4.2. LED Supraharmonic Emission Test

This test measures an LED lamp with a switching frequency of approximately 100 kHz and a power of 30 watts, using an AC laboratory power supply to power it. As shown in Figure 8, a voltage probe and a current probe are connected at the connection point between the power supply and the LED lamp. The voltage and current waveforms of the LED lamp obtained from the test are shown in Figure 9.

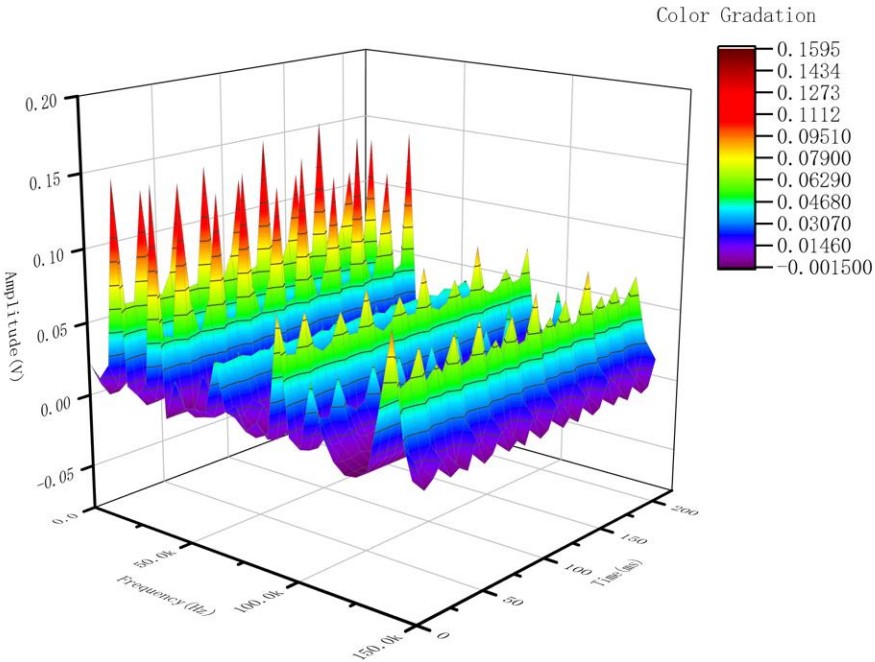

**Figure 7.** Class D signal test experimental results.

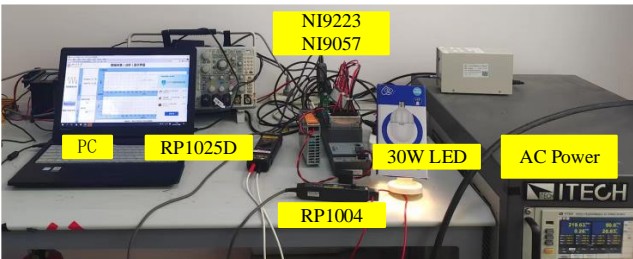

**Figure 8.** LED test settings.

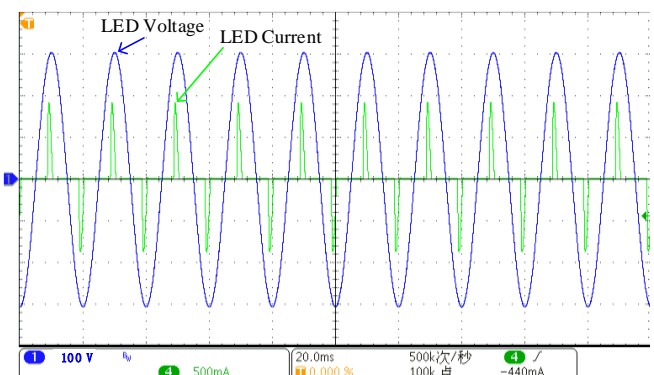

**Figure 9.** LED voltage and current waveforms.

A simplified compressed sensing model with multiple measurement vectors is used for data processing, while 400 sets of data of a 200 ms test duration are calculated, the algorithm interpolation factor is set to 10, and the sparsity is predicted using a probability density distribution model. The sparsity of the voltage spectrum was predicted to be eight. The voltage detection results are shown in Figure 10a, and the supraharmonic emission is concentrated in the 2–10 kHz band and the switching frequency, where the instantaneous amplitude of the detection results at 2–10 kHz is large but not regular, while the amplitude detection results at the switching frequency have obvious periodicity, and

the period is related to the fundamental frequency. The sparsity of the current spectrum was predicted to be six, and the current detection results are shown in Figure 10b, where the current amplitude period variation at the switching frequency is still correlated with the fundamental frequency period.

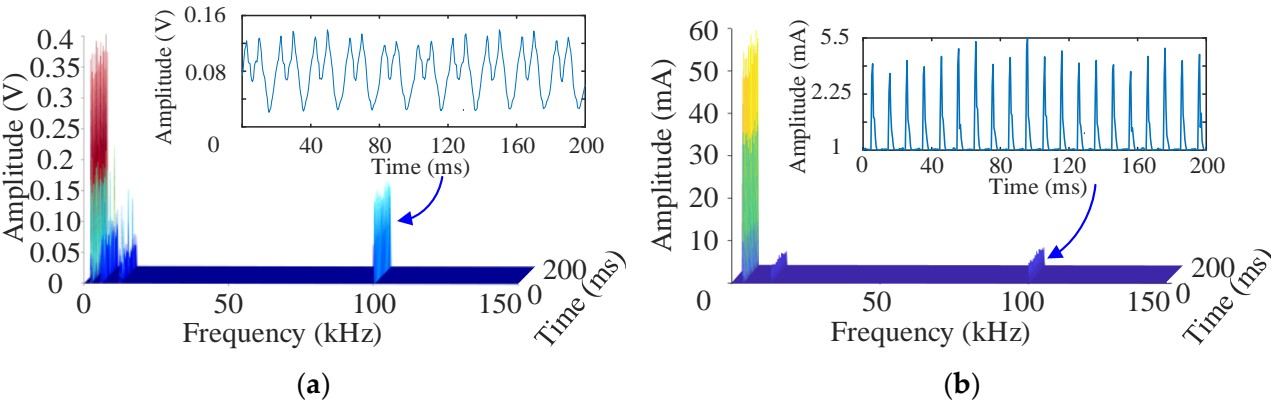

**Figure 10.** (**a**) LED voltage time and frequency diagram; (**b**) LED current time frequency diagram.

### 4.3. Experimental Time Comparison

After conducting the experiments for LED and Class D test signals, the computation time is summarized in Table 6. The calculated times in the table are the average of 200 experimental times. The sparsity of the MCS-OMP algorithm is set to the sparsity predicted by the probabilistic model in the table to compare the computation time of the simplified compressed sensing model. In the LED lamp experiments, the voltage spectrum computation time of the algorithm in this paper was 36 ms, and the current spectrum computation time was 34 ms, while the voltage spectrum computation time of the MCS-OMP algorithm was 155 ms, and the current spectrum computation time was 132 ms.

**Table 6.** Calculation time comparison.

| Algorithm | Time(ms) | | |
|---|---|---|---|
| | LED Voltage | LED Current | Class D |
| New proposal | 35.7 | 34.2 | 106.3 |
| MCS-OMP | 155.1 | 132.3 | 387.4 |

In the Class D signal experiment, the voltage spectrum calculation time was 106 ms, and the voltage spectrum calculation time of the MCS-OMP algorithm was 386 ms. Therefore, if the sparsity estimation in online applications is large, the calculation time of the MCS-OMP algorithm may exceed 200 ms, and the algorithm in this paper can shorten the time on the basis of MCS-OMP, further improving the online detection of supraharmonic feasibility.

### 5. Conclusions

In this paper, a new measurement method for identifying and estimating the supraharmonic components is introduced and validated. The method first estimates the sparsity of the original spectral array and the index of the main supraharmonic emission, which is used to further simplify the existing MMV measurement model, and, finally, recovers the high-resolution spectral array based on CS-OMP. In Section 3, two simulation models were used to verify the performance of the algorithm. Compared with existing compressed sensing algorithms, the algorithm in this paper achieves significant time reduction with guaranteed computational accuracy, followed by the further verification of the algorithm's effectiveness in estimating the sparsity of practical applications with the measurement data in Section 4 and the computational results using the MMV model are shown.

The simulation and experimental results confirm that the method effectively reduces computational complexity while maintaining the accuracy of the compressive sensing estimation. Meanwhile, the method in this paper can be further applied to other existing compression-aware algorithms to shorten the computation time in order to reduce the online measurement time. The future steps of this study are to shorten the computation time based on improving the frequency resolution of the supraharmonic test.

**Author Contributions:** Data curation, H.Z.; Methodology, Z.G.; Supervision, Z.L.; Validation, Q.Z.; Visualization, Z.W.; Writing—original draft, Z.G.; Writing—review & editing, Z.G. and Q.Z. All authors have read and agreed to the published version of the manuscript.

**Funding:** This research received no external funding.

**Conflicts of Interest:** The authors declare no conflict of interest.

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
