# Peer review of "Fast Supraharmonic Estimation Algorithm Based on Simplified Compressed Sensing Model"

_electronics, doi:10.3390/electronics12010141_

Round 1
Reviewer 1 Report
After reading the article, I’d like to pay attention to the following comments.
1) The authors show the graphic (see the Fig.3) to prove the performance growth, on which we see that the algorithm computation time has nonlinear growth behaviour. Therefore, in the situation of sparsity increase, the execution time of author’s algorithm will go beyond the execution time periods of other algorithms. It is necessary to ground why the considered range of sparsity is sufficient.
2) The authors place no requirements to the accuracy and the target performance. Obviously, these will be application restriction factors. A task should be selected/determined which will greatly demonstrate the advantages of the proposed approach.
3) The use of spectral decomposition is a well-known technique. It is necessary to highlight its distinctive feature.
4) We all know such techniques as power spectral density and periodograms which are applied to solve conceptually close tasks (see the example – DOI: 10.25673/76935). The author’s approach is conceptually close to this one, however, the authors do not explain why they denied to use the mathematical methods.
5) Since the authors performed calculations with help of Windows system (which doesn’t provide real time system), it is suggested to perform multiple calculations and to calculate the dispersion value by estimating the computation time value (see https://en.wikipedia.org/wiki/Central_limit_theorem).
I wish the authors good luck and a quick publication!
Author Response
Thanks very much for your valuable comments. We sincerely wish you good health and all the best. Please see the attachment.

Reviewer 2 Report
This paper considers supraharmonic estimation algorithm based on compressed sensing. It aims to reduce the sparsity in advance to simplify the recovery process. However, the proposed method seems intractable and less motivated. First, CS-OMP can be realized by minimizing the sparsity subject to a desired residual energy. It will not necessarily require the sparsity in this situation. Then, the proposed sparsity estimation method is unclear and unconvinced. Finally, reducing the size of $D$ from N-by N‘ to N-by FS is confusing since we cannot generally change the dimention of recovered signal in a general CS problem. If FS < N, it turns to a simple overdetermined optimization problem.
Author Response

(The authors gave the same response as above.)

Reviewer 3 Report
Dear authors
The authors created a new method to improve online applications.
Their simulation and experimental results confirm that the method effectively reduces the computational complexity while maintaining the accuracy of the compressive sensing estimation.
Their simulation verifies that the model can reduce the computation time to less than half of the original compressed sensing model and does not affect the computation accuracy.
Some typos in English must be corrected like
above mentioned-->mentioned above
high frequency-->high-frequency
device life-->device's life
[8]. the meter-->[8]. The meter
also affect-->also affects
Table III--->Table 3
And others
Regards
Author Response

(The authors gave the same response as above.)

Round 2
Reviewer 2 Report
I have no more comments